# DeepSFM: Structure From Motion Via Deep Bundle Adjustment

## Abstract

Structure from motion (SfM) is an essential computer vision problem which has not been well handled by deep learning. One of the promising trends is to apply explicit structural constraint, e.g. 3D cost volume, into the network. In this work, we design a physical driven architecture, namely DeepSFM, inspired by traditional Bundle Adjustment (BA), which consists of two cost volume based architectures for depth and pose estimation respectively, iteratively running to improve both. In each cost volume, we encode not only photo-metric consistency across multiple input images, but also geometric consistency to ensure that depths from multiple views agree with each other. The explicit constraints on both depth (structure) and pose (motion), when combined with the learning components, bring the merit from both traditional BA and emerging deep learning technology. Extensive experiments on various datasets show that our model achieves the state-of-the-art performance on both depth and pose estimation with superior robustness against less number of inputs and the noise in initialization.

## 1 Introduction

Structure from motion (SfM) is a fundamental human vision functionality which recovers 3D structures from the projected retinal images of moving objects or scenes. It enables machines to sense and understand with the 3D world and is critical in achieving real-world artificial intelligence. Over decades of researches, there has been a lot of great success on SfM; however, the performance is far from perfect.

Conventional SfM approaches (Agarwal et al., 2011; Wu et al., 2011a; Engel et al., 2017; Delaunoy & Pollefeys, 2014) heavily rely on Bundle-Adjustment (BA) (Triggs et al., 1999; Agarwal et al., 2010), in which 3D structures and camera motions of each view are jointly optimized via Levenberg-Marquardt (LM) algorithm (Nocedal & Wright, 2006) according to the cross-view correspondence. Though successful in certain scenarios, conventional SfM based approaches are fundamentally restricted by the coverage of the provided multiple views and the overlaps among them. They also typically fail to reconstruct textureless or non-lambertian (e.g. reflective or transparent) surfaces due to the missing of correspondence across views. As a result, selecting sufficiently good input views and the right scene requires excessive caution and is usually non-trivial to even experienced user.

Recent researches resort to deep learning to deal with the typical weakness of conventional SfM. Early effort utilizes deep neural network as a powerful mapping function that directly regresses the structures and motions (Ummenhofer et al., 2017; Vijayanarasimhan et al., 2017; Zhou et al., 2017; Wang et al., 2017). Since the geometric constraints of structures and motions are not explicitly enforced, the network does not learn the underlying physics and prone to overfitting. Consequently, they do not perform as accurate as conventional SfM approaches and suffer from extremely poor generalization capability. Most recently, the 3D cost volume (Teed & Deng, 2018) has been introduced to explicit leveraging photo-consistency in a differentiable way, which significantly boosts the performance of deep learning based 3D reconstruction. However, the camera motion usually has to be known (Yao et al., 2018; Im et al., 2019) or predicted via direct regression (Ummenhofer et al., 2017; Zhou et al., 2017; Teed & Deng, 2018), which still suffer from generalization issue.

In this paper, we explicitly enforce photo-consistency, geometric-consistency, and camera motion constraints in a unified deep learning framework. In particular, our network includes a depth based cost volume (D-CV) and a pose based cost volume (P-CV). D-CV optimizes per-pixel depth values

with the current camera poses, while P-CV optimizes camera poses with the current depth estimations. Conventional 3D cost volume enforces photo-consistency by unprojecting pixels into the discrete camera fronto-parallel planes and computing the photometric (i.e. image feature) difference as the cost. In addition to that, our D-CV further enforces geometric-consistency among cameras with their current depth estimations by adding the geometric (i.e. depth) difference to the cost. Note that the initial depth estimation can be obtained using the conventional 3D cost volume. For pose estimation, rather than direct regression, our P-CV discretizes around the current camera positions, and also computes the photometric and/or geometric differences by hypothetically moving the camera into the discretized position. Note that the initial camera pose can be obtained by a rough estimation from the direct regression methods such as (Ummenhofer et al., 2017). Our framework bridges the gap between the conventional and deep learning based SfM by incorporating explicit constraints of photo-consistency, geometric-consistency and camera motions all in the deep network.

The closest work in the literature is the recently proposed BA-Net (Tang & Tan, 2018), which also aims to explicitly incorporate multi-view geometric constraints in a deep learning framework. They achieve this goal by integrating the LM optimization into the network. However, the LM iterations are unrolled with few iterations due to the memory and computational inefficiency, and thus it may lead to non-optimal solutions. In contrast, our method does not have a restriction on the number of iterations and achieves empirically better performance. Furthermore, LM in SfM originally optimizes point and camera positions, and thus direct integration of LM still requires good correspondences. To evade the correspondence issue in typical SfM, their models employ a direct regressor to predict depth at the front end, which heavily relies on prior in the training data. In contrast, our model is a fully physical-driven architecture that less suffers from over-fitting issue for both depth and pose estimation.

To demonstrate the superiority of our method, we conduct extensive experiments on DeMoN datasets, ScanNet and ETH3D. The experiments show that our approach outperforms the state-of-the-art Schonberger & Frahm (2016); Ummenhofer et al. (2017); Tang & Tan (2018).

## 2 RELATED WORK

There is a large body of work that focuses on inferring depth or motion from color images, ranging from single view, multiple views and monocular video. We discuss them in the context of our work.

**Single-view Depth Estimation.** While ill-posed, the emerging of deep learning technology enables the estimation of depth from a single color image. The early work directly formulates this into a per-pixel regression problem (Eigen et al., 2014), and follow-up works improve the performance by introducing multi-scale network architectures (Eigen et al., 2014; Eigen & Fergus, 2015), skip-connections (Wang et al., 2015; Liu et al., 2016), powerful decoder and post process (Garg et al., 2016; Laina et al., 2016; Kuznietsov et al., 2017; Wang et al., 2015; Liu et al., 2016), and new loss functions (Fu et al., 2018). Even though single view based methods generate plausible results, the models usually resort heavily to the prior in the training data and suffer from generalization capability. Nevertheless, these methods still act as an important component in some multi-view systems (Tang & Tan, 2018)

**Traditional Structure-from-Motion** Simultaneously estimating 3d structure and camera motion is a well studied problem which has a traditional tool-chain of techniques (Furukawa et al., 2010; Newcombe et al., 2011; Wu et al., 2011b). Structure from Motion(SfM) has made great progress in many aspects. Lowe (2004); Han et al. (2015) aim at improving features and Snavely (2011) introduce new optimization techniques. More robust structures and data representations are introduced by Gherardi et al. (2010); Schonberger & Frahm (2016). Simultaneous Localization and Sapping(SLAM) systems track the motion of the camera and build 3D structure from video sequence (Newcombe et al., 2011; Engel et al., 2014; Mur-Artal et al., 2015; Mur-Artal & Tardós, 2017). Engel et al. (2014) propose the photometric bundle adjustment algorithm to directly minimize the photometric error of aligned pixels. However, traditional SfM and SLAM methods are sensitive to low texture region, occlusions, moving objects and lighting changes, which limit the performance and stability.

**Deep Learning for Structure-from-Motion** Deep neural networks have shown great success in stereo matching and Structure-from-Motion problems. Ummenhofer et al. (2017); Wang et al. (2017); Vijayanarasimhan et al. (2017); Zhou et al. (2017) regress depth map and camera pose di-

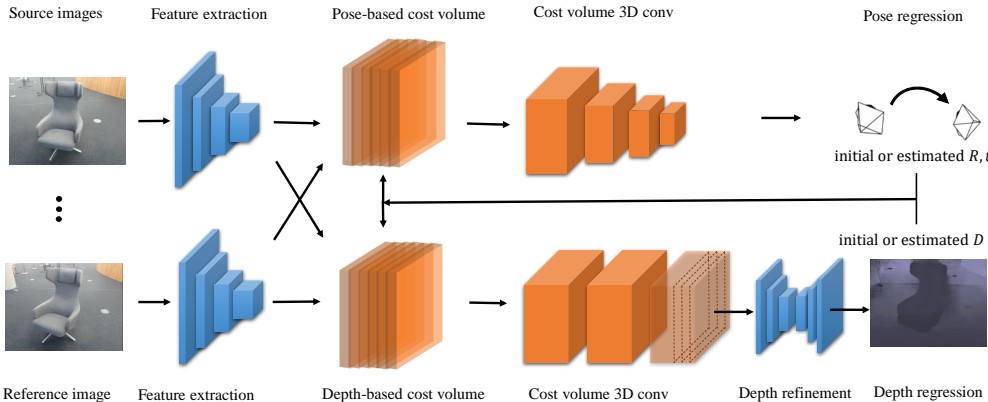

Figure 1: Overview of our method. 2D CNN is used to extract photometric feature to construct cost volumes. Initial source depth maps are used to introduce geometry consistency. A series of 3D CNN layers are applied for both pose based cost volume and D-CV. Then a context network and depth regression operation are applied to produce predicted depth map of reference image.

rectly in a supervised manner or by introducing photometric constraints between depth and motion as a self-supervision signal. Such methods solve the camera motion as a regression problem, and the relation between camera motion and depth prediction is neglected.

Recently, some methods exploit multi-view photometric or feature-metric constraints to enforce the relationship between dense depth map and the camera pose in network. The SE3 transformer layer is introduced by Teed & Deng (2018), which uses geometry to map flow and depth into a camera pose update. Wang et al. (2018) propose the differentiable camera motion estimator based on the Direct Visual Odometry (Steinbrücker et al., 2011). Clark et al. (2018) using a LSTM-RNN (Hochreiter et al., 2001) as the optimizer to solve nonlinear least squares in two-view SfM. Tang & Tan (2018) train a network to generate a set of basis depth maps and optimize depth and camera poses in a BA-layer by minimizing a feature-metric error.

## 3 ARCHITECTURE

Our framework receives frames of a scene from different viewpoints, and produces photo-metrically and geometrically consistent depth maps across all frames and the corresponding camera poses. Similar to BA, we also assume initial structures (i.e depth maps) and motions (i.e. camera poses) are given. Note that the initialization is not necessary to be super accurate for the good performance using our framework and thus can be easily obtained from some direct regression methods (Ummenhofer et al., 2017).

Now we introduce the detail of our model – DeepSFM. Without loss of generality, we describe our model taking two images as input, namely the reference image and the source image, as an example, and all the technical components can be extended for multiple images straightforward. As shown in Figure 1, we first extract feature maps from input images through a shared encoder. We then sample the solution space for depth uniformly in the inverse-depth space between a predefined minimum and maximum range and camera pose around the initialization respectively. After that, we build cost volumes accordingly to reason the confidence of each hypothesis. This is achieved by validating the consistency between the feature of the reference view and the ones warped from the source image. Besides photo-metric consistency that measures the color image similarity, we also take into account the geometric consistency across warped depth maps. Note that depth and pose require different designs of cost volume to efficiently sample the hypothesis space. Gradients can back-propagate through cost volumes, and cost-volume construction does not affect any trainable parameters. The cost volumes are then fed into 3D CNN to regress new depth and pose. These updated value can be used to create new cost volumes, and the model improves the prediction iteratively.

For notations, we denote $\{\mathbf{I}_i\}_{i=1}^n$ as the image sequences in one scene, $\{\mathbf{D}_i\}_{i=1}^n$ as the corresponding ground truth depth maps, $\{\mathbf{K}_i\}_{i=1}^n$ as the camera intrinsics, $\{\mathbf{R}_i, \mathbf{t}_i\}_{i=1}^n$ as the ground truth rotations

and translations of camera, $\{\mathbf{D}_i^*\}_{i=1}^n$ and $\{\mathbf{R}_i^*, \mathbf{t}_i^*\}_{i=1}^n$ as initial depth maps and camera pose parameters for constructing cost volumes, where $n$ is the number of image samples.

## 3.1 2D Feature Extraction

Given the input sequences $\{\mathbf{I}_i\}_{i=1}^n$, we extract the 2D CNN feature $\{\mathbf{F}_i\}_{i=1}^n$ for each frame. Firstly, a 7 layers' CNN with kernel size $3 \times 3$ is applied to extract low contextual information. Then we adopt a spatial pyramid pooling (SPP) (Kaiming et al., 2014) module, which can extract hierarchical multi-scale features through 4 average pooling blocks with different pooling kernel size ($4 \times 4, 8 \times 8, 16 \times 16, 32 \times 32$). Finally, we pass the concatenated features through 2D CNNs to get the 32-channel image features after upsampling these multi-scale features into the same resolution. These image sequence features are used by the building of both our depth based and pose based cost volumes.

## 3.2 Depth based Cost Volume (D-CV)

Traditional plane sweep cost volume aims to back-project the source images onto successive virtual planes in the 3D space and measure photo-consistency error among the warped image features and reference image features for each pixel. Different from the cost volume used in previous multi-view and structure-from-motion methods, we construct a D-CV to further utilize the local geometric consistency constraints introduced by depth maps. Inspired by the traditional plane sweep cost volumes, our D-CV is a concatenation of three components: the reference image features, the warped source image features and the homogeneous depth consistency maps.

**Hypothesis Sampling** To back-project the features and depth maps from source viewpoint to the 3D space in reference viewpoint, we uniformly sample a set of $L$ virtual planes $\{d_l\}_{l=1}^L$ in the inverse-depth space which are perpendicular to the forward direction ($z$-axis) of the reference viewpoint. These planes serve as the hypothesis of the output depth map, and the cost volume can be built upon them.

**Feature warping.** To construct our D-CV, we first warp source image features $\mathbf{F}_i$ (of size $CHannel \times Width \times Height$) to each of the hypothetical depth map planes $d_l$ using camera intrinsic matrix $\mathbf{K}$ and initial camera poses $\{\mathbf{R}_i^*, \mathbf{t}_i^*\}$, according to:

$$\tilde{\mathbf{F}}_{il}(u) = \mathbf{F}_i(\tilde{u}_l), \tilde{u}_l \sim \mathbf{K}[\mathbf{R}_i^*|\mathbf{t}_i^*]\begin{bmatrix}(\mathbf{K}^{-1}u)d_l \\ 1\end{bmatrix} \tag{1}$$

where $u$ and $\tilde{u}_l$ are the homogeneous coordinates of each pixel in the reference view and the projected coordinates onto the corresponding source view. $\tilde{\mathbf{F}}_{il}(u)$ denotes the warped feature of the source image through the $l$-th virtual depth plane. Note that the projected homogeneous coordinates $\tilde{u}_l$ are floating numbers, and we adopt a differentiable bilinear interpolation to generate the warped feature map $\tilde{\mathbf{F}}_{il}$. The pixels with no source view coverage are assigned with zeros. Following Im et al. (2019), we concatenate the reference feature and the warped reference feature together and obtain a $2CH \times L \times W \times H$ 4D feature volume.

**Depth consistency.** In addition to photometric consistency, to exploit geometric consistency and promote the quality of depth prediction, we add two more channels on each virtual plane: the warped initial depth maps from the source view and the depth map of the virtual plane from the perspective of the source view. Note that the former is the same as image feature warping, while the latter requires a coordinate transformation from the reference camera to the source camera.

In particular, the first channel is computed as follows. The initial depth map of source image is first down-sampled and then warped to hypothetical depth planes based on initial camera pose similarly to the image feature warping:

$$\tilde{\mathbf{D}}_{il}^*(u) = \mathbf{D}_i^*(\tilde{u}_l) \tag{2}$$

where the coordinates $u$ and $\tilde{u}_l$ are defined in Eq. 1 and $\tilde{\mathbf{D}}_{il}^*(u)$ represents the warped one-channel depth map on the $l$-th depth plane. One distinction between depth warping and feature warping is that we adopt nearest neighbor sampling for depth warping, instead of bilinear interpolation. A comparison between the two methods are provided in Appendix C.

The second channel contains the depth values of the virtual planes in the reference view by seeing them from the source view. To transform the virtual planes to the source view coordinate system, we apply a $T$ function on each virtual plane $d_l$ in the following:

$$T(d_l) \sim [\mathbf{R}_i^*|\mathbf{t}_i^*] \left[ \begin{array}{c} \left(\mathbf{K}^{-1}u\right) d_l \\ 1 \end{array} \right] \tag{3}$$

We stack the warped initial depth maps and the transformed depth planes together, and get a depth volume of size $2 \times L \times W \times H$.

By concatenating the feature volume and depth volume together, we obtain a 4D cost tensor of size $(2CH + 2) \times L \times W \times H$. Given the 4D cost volume, our network learns a cost volume of size $L \times W \times H$ using several 3D convolutional layers with kernel size $3 \times 3 \times 3$. When there is more than one source image, we get the final cost volume by averaging over multiple input source views.

### 3.3 POSE BASED COST VOLUME (P-CV)

In addition to the construction of D-CV, we also propose a P-CV, aiming at optimizing initial camera poses through both photometric and geometric consistency. Instead of building a cost volume based on hypothetical depth map planes, our novel P-CV is constructed based on a set of assumptive camera poses. Similar to D-CV, P-CV is also concatenated by three components: the reference image features, the warped source image features and the homogeneous depth consistency maps. Given initial camera pose parameters $\{\mathbf{R}_i^*, \mathbf{t}_i^*\}$, we uniformly sample a batch of discrete candidate camera poses around. Since jointly sampling camera rotation and translation along 6-DoF is costly, we shift rotation and translation separately by keeping one frozen while sampling the other one. In the end, a group of $P$ virtual camera poses noted as $\{\mathbf{R}_{ip}^*|\mathbf{t}_{ip}^*\}_{p=1}^P$ around input pose are obtained for cost volume construction.

The posed-based cost volume is also constructed by concatenating image features and homogeneous depth maps. However, source view features and depth maps are warped based on sampled camera poses. For feature warping, we compute $\tilde{u}_p$ as following equations:

$$\tilde{u}_p \sim \mathbf{K} \left[ \mathbf{R}_{ip}^*|\mathbf{t}_{ip}^* \right] \left[ \begin{array}{c} \left(\mathbf{K}^{-1}u\right) \mathbf{D}_i^* \\ 1 \end{array} \right] \tag{4}$$

where $\mathbf{D}_i^*$ is the initial reference view depth. Similar to D-CV, we get warped source feature map $\tilde{\mathbf{F}}_{ip}$ after bilinear sampling and concatenate it with reference view feature map. We also transform the initial reference view depth and source view depth into one homogeneous coordinate system, which enhances the geometric consistency between camera pose and multi view depth maps.

After concatenating the above feature maps and depth maps together, we again build a 4D cost volume of size $(2CH + 2) \times P \times W \times H$, where $W$ and $H$ are the width and height of feature map, $CH$ is the number of channels. We get output of size $1 \times P \times 1 \times 1$ from the above 4-D tensor after eight 3D convolutional layers with kernel size $3 \times 3 \times 3$, three 3D average pooling layers with stride size $2 \times 2 \times 1$ and one global average pooling at the end.

### 3.4 COST AGGREGATION AND REGRESSION

For depth prediction, we follow the cost aggregation technique introduced by Im et al. (2019). We adopt a context network, which takes reference image features and each slice of the coarse cost volume after 3D convolution as input and produce the refined cost slice. The final aggregated depth based volume is obtained by adding coarse and refined cost slices together. The last step to get depth prediction of reference image is depth regression. We pass each slice of D-CV through a soft-max function to get the probability of every depth value $l$. Then the weighted sum of all hypothetical depth values is regarded as predicted depth map; this operation is called soft-argmax. We can also get the predicted coarse depth map by the same way using coarse D-CV. For camera poses prediction, we also apply a soft-argmax function on pose cost volume and get the estimated output rotation and translation vectors.

## 3.5 Training

The DeepSFM learns the feature extractor, cost aggregation, and the regression layers in a supervised way. We denote $\hat{\mathbf{R}}_i$ and $\hat{\mathbf{t}}_i$ as predicted rotation angles and translation vectors of camera pose. Then the pose loss function is defined as the $L1$ distance between prediction and groundtruth: $\mathcal{L}_{rotation} = \left|\hat{\mathbf{R}}_i - \mathbf{R}_i\right|$ and $\mathcal{L}_{translation} = \left|\hat{\mathbf{t}}_i - \mathbf{t}_i\right|$. We denote $\hat{D}_i^0$ and $\hat{D}_i$ as predicted coarse depth map and refined depth map for the $i$-th image, then the depth loss function is defined as following equation:

$$\mathcal{L}_{depth} = \sum_i \lambda H(\hat{D}_i^0, \mathbf{D}_i) + H(\hat{D}_i, \mathbf{D}_i) \tag{5}$$

where $\lambda$ is weight parameter and function $H$ is Huber loss.

Our final objective becomes

$$\mathcal{L}_{final} = \lambda_r \mathcal{L}_{rotation} + \lambda_t \mathcal{L}_{translation} + \lambda_d \mathcal{L}_{depth} \tag{6}$$

We follow two rules to set $\lambda_r$, $\lambda_t$ and $\lambda_d$: 1) the loss term provides gradient on the same order of numerical value range, such that no single loss term could dominate the training process, since accuracy in depth and camera pose are both important to reach a good consensus. 2) we found in practice the camera rotation has higher impact on the accuracy of the depth but not the opposite. To encourage better performance of pose, we set a relatively large $\lambda_r$. In practice, the weight parameter $\lambda$ to balance loss objective is set to 0.7, while $\lambda_r = 0.8$, $\lambda_t = 0.1$ and $\lambda_d = 0.1$.

The RGB sequences, corresponding ground-truth depth maps and camera intrinsics and extrinsics are fed as input samples. We initialize the 2D feature extraction layers with pre-trained DPSNet weight. The initial depth maps and camera poses $\{\mathbf{D}_i^*\}_{i=1}^n$ and $\{\mathbf{R}_i^*, \mathbf{t}_i^*\}_{i=1}^n$ are obtained from De-MoN. To keep correct scale, we multiply translation vectors and depth maps by the norm of the ground truth camera translation vector. The whole training and testing procedure are performed as four iterations. During each iteration, we take the predicted depth maps and camera poses of previous iteration as new $\{\mathbf{D}_i^*\}_{i=1}^n$ and $\{\mathbf{R}_i^*, \mathbf{t}_i^*\}_{i=1}^n$ for cost volume construction.

We implement our system using PyTorch framework. The training procedure takes 6 days on 3 NVIDIA TITAN GPUs on all 160k training sequences. The training batch size is set to 4, and the Adam optimizer ($\beta_1 = 0.9, \beta_2 = 0.999$) is used with learning rate $2 \times 10^{-4}$, which decreases to $4 \times 10^{-5}$ after 2 epochs. Within the first two epochs, the parameters in 2D CNN feature extraction module are frozen, and the ground truth depth maps for source images are used to construct D-CV and P-CV, which are replaced with predicted depth maps from network in latter epochs. During training process, the length of input sequences is 2 (one reference image and one source image). The $L$ for D-CV is set to 64 and the N for P-CV is 10. The range of both cost volumes is adapted during training and testing.

## 4 Experiments

### 4.1 Datasets

We evaluate DeepSFM on widely used datasets and compare to state-of-the-art methods on accuracy and generalization capability.

**DeMoN Datasets** Proposed in DeMoN (Ummenhofer et al., 2017), this dataset contains data from various sources, including SUN3D (Xiao et al., 2013), RGB-D SLAM (Sturm et al., 2012), and Scenes11 (Chang et al., 2015). To test the generalization capability, we also evaluate on MVS (Fuhrmann et al., 2014) dataset but not use it for the training. In all four datasets, RGB image sequences and the ground truth depth maps are provided with the camera intrinsics and camera poses. Note that those datasets together provide a diverse set of both indoor and outdoor, synthetic and real-world scenes. Specifically, Scenes11 consists of synthetic images rendered from random scenes, on which ground truth camera poses and depth are perfect, but objects are lack of reality in scale and semantics. For training and testing, we use the same setting as DeMoN.

**ETH3D Dataset** ETH3D dataset provides a variety of indoor and outdoor scenes with high-precision ground truth 3D points captured by laser scanners, which is a more solid benchmark

dataset. Ground truth depth maps are obtained by projecting the point clouds to each camera view. Raw images are in high resolution but resized to $810 \times 540$ pixels for evaluation due to memory constraint. Again, all the models are trained on DeMoN and tested here.

| **MVS** | Depth | | | Motion | | **Scenes11** | Depth | | | Motion | |
|---|---|---|---|---|---|---|---|---|---|---|---|
| Method | L1-inv | sc-inv | L1-rel | Rot | Trans | Method | L1-inv | sc-inv | L1-rel | Rot | Trans |
| Base-Oracle | 0.019 | 0.197 | 0.105 | 0 | 0 | Base-Oracle | 0.023 | 0.618 | 0.349 | 0 | 0 |
| Base-SIFT | 0.056 | 0.309 | 0.361 | 21.180 | 60.516 | Base-SIFT | 0.051 | 0.900 | 1.027 | 6.179 | 56.650 |
| Base-FF | 0.055 | 0.308 | 0.322 | 4.834 | 17.252 | Base-FF | 0.038 | 0.793 | 0.776 | 1.309 | 19.426 |
| Base-Matlab | - | - | - | 10.843 | 32.736 | Base-Matlab | - | - | - | 0.917 | 14.639 |
| DeMoN | 0.047 | 0.202 | 0.305 | 5.156 | 14.447 | DeMoN | 0.019 | 0.315 | 0.248 | 0.809 | 8.918 |
| LS-Net | 0.051 | 0.221 | 0.311 | 4.653 | 11.221 | LS-Net | 0.010 | 0.410 | 0.210 | 4.653 | 8.210 |
| BANet | 0.030 | 0.150 | 0.080 | 3.499 | 11.238 | BANet | 0.080 | 0.210 | 0.130 | 3.499 | 10.370 |
| Ours | **0.021** | **0.129** | **0.079** | **2.824** | **9.881** | Ours | **0.007** | **0.112** | **0.064** | **0.403** | **5.828** |
| **RGB-D** | Depth | | | Motion | | **Sun3D** | Depth | | | Motion | |
| Method | L1-inv | sc-inv | L1-rel | Rot | Trans | Method | L1-inv | sc-inv | L1-rel | Rot | Trans |
| Base-Oracle | 0.026 | 0.398 | 0.36 | 0 | 0 | Base-Oracle | 0.020 | 0.241 | 0.220 | 0 | 0 |
| Base-SIFT | 0.050 | 0.577 | 0.703 | 12.010 | 56.021 | Base-SIFT | 0.029 | 0.290 | 0.286 | 7.702 | 41.825 |
| Base-FF | 0.045 | 0.548 | 0.613 | 4.709 | 46.058 | Base-FF | 0.029 | 0.284 | 0.297 | 3.681 | 33.301 |
| Base-Matlab | - | - | - | 12.813 | 49.612 | Base-Matlab | - | - | - | 5.920 | 32.298 |
| DeMoN | 0.028 | 0.130 | 0.212 | 2.641 | 20.585 | DeMoN | 0.019 | 0.114 | 0.172 | 1.801 | 18.811 |
| LS-Net | 0.019 | 0.090 | 0.301 | **1.010** | 22.100 | LS-Net | 0.015 | 0.189 | 0.650 | **1.521** | 14.347 |
| BANet | **0.008** | 0.087 | **0.050** | 2.459 | 14.900 | BANet | 0.015 | 0.110 | **0.060** | 1.729 | 13.260 |
| Ours | 0.011 | **0.071** | 0.126 | 1.862 | **14.570** | Ours | **0.013** | **0.093** | 0.072 | 1.704 | **13.107** |

Table 1: Results on MVS, SUN3D, RGBD and Scenes11, the best results are noted by **Bold**.

## 4.2 EVALUATION

**DeMoN Datasets** Our results on DeMoN datasets and the comparison to other methods are shown in Table 1. We cite results of some strong baseline methods from DeMoN paper, named as Base-Oracle, Base-SIFT, Base-FF and Base-Matlab respectively (Ummenhofer et al., 2017). Base-Oracle estimate depth with the ground truth camera motion using SGM (Hirschmuller, 2005). Base-SIFT, Base-FF and Base-Matlab solve camera motion and depth using feature, optical flow, and KLT tracking correspondence from 8-pt algorithm (Hartley, 1997). We also compare to some most recent state-of-the-art methods LS-Net (Clark et al., 2018) and BA-Net (Tang & Tan, 2018). LS-Net introduces the learned LSTM-RNN optimizer to minimizing photometric error for stereo reconstruction. BA-Net is the most recent work that minimizes the feature-metric error between multi-view via the differentiable Levenberg-Marquardt (Lourakis & Argyros, 2005) algorithm.

To make a fair comparison, we adopt the same error metrics as DeMoN for depth and camera pose evaluation. L1-inv computes the disparity map errors, and sc-inv is a scale-invariant error metric.

| Method | Error metric | | | | | Accuracy metric($\delta < \alpha^t$) | | |
|---|---|---|---|---|---|---|---|---|
| | abs_rel | abs_diff | sq_rel | rms | log_rms | $\alpha$ | $\alpha^2$ | $\alpha^3$ |
| COLMAP | 0.324 | **0.615** | 36.71 | 2.370 | 0.349 | **86.5** | 90.3 | 92.7 |
| DeMoN | 0.191 | 0.726 | 0.365 | 1.059 | 0.240 | 73.3 | 89.8 | 95.1 |
| Ours | **0.127** | 0.661 | **0.278** | **1.003** | **0.195** | 84.1 | **93.8** | **96.9** |

Table 2: Results on ETH3D (**Bold**: best; $\alpha = 1.25$). abs_rel, abs_diff, sq_rel, rms, and log_rms, are absolute relative error, absolute difference, square relative difference, root mean square and log root mean square, respectively.

L1-rel measures the depth errors relative to the ground truth depth, which emphasize depth estimation of close range in the scene. For camera poses evaluation, the angles between the prediction and the ground truth rotation and translation are shown as Rot and Trans respectively.

Our method outperforms all traditional baseline methods and DeMoN on both depth and camera poses. When compared to more recent LS-Net and BA-Net, our method produces better results in most metrics of the four datasets. On RGB-D dataset, our performance is comparable to the state-of-the-art due to relatively higher noise in the RGB-D ground truth. LS-Net trains an initialization network which regresses depth and motion directly before adding the LSTM-RNN optimizer. The performance of the RNN optimizer is highly affected by the accuracy of the regressed initialization. The depth results of LS-Net are consistently poorer than BA-Net and our method, despite better rotation parameters are estimated by LS-Net on RGB-D and Sun3D datasets with very good initialization. Our method is slightly inferior to BA-Net on the L1-rel metric, which is probably due to that we sample 64 virtual planes uniformly as the hypothetical depth set, while BA-Net optimizes depth prediction based on a set of 128-channel estimated basis depth maps that are more memory consuming but have more fine-grained results empirically. Despite all that, it is shown that our learned cost volumes with geometric consistency work better than the photometric bundle adjustment (e.g. used in BA-Net) in most scenes. In particular, we improve mostly on the Scenes11 dataset, where the ground truth is perfect but the input images contain a lot of texture-less regions, which are challenging to photo-consistency based methods.

**ETH3D** We further test the generalization capability on ETH3D. We provide comparisons to COLMAP (Schonberger & Frahm, 2016) and DeMoN on ETH3D. COLMAP is a state-of-the-art Structure-from-Motion method, while DeMoN introduces a classical deep network architecture that directly regress depth and motion in a supervised manner. In the accuracy metric, the error $\delta$ s defined as $\max(\frac{y_i^*}{y_i}, \frac{y_i}{y_i^*})$, and the thresholds are typically set as $[1.25, 1.25^2, 1.25^3]$. In Table 2, our method shows the best performance overall among all the comparison methods. Our method produces better results than DeMoN consistently, since we impose geometric and physical constraints onto network rather than learning to regress directly. When compared with COLMAP, our method performs better on most metrics. COLMAP behaves well in the accuracy metric (i.e. abs_diff). However, the presence of outliers is often observed in the predictions of COLMAP, which leads to poor performance in other metrics such as abs_rel and sq_rel, since those metrics are sensitive to outliers. We put more qualitative comparisons with COLMAP in Appendix C. For more comparison on generalization, another experiment on ScanNet is provided in Appendix B.

### 4.3 MODEL ANALYSIS

In this section, we analyze our model on several aspects to verify the optimality and show advantages over previous methods.

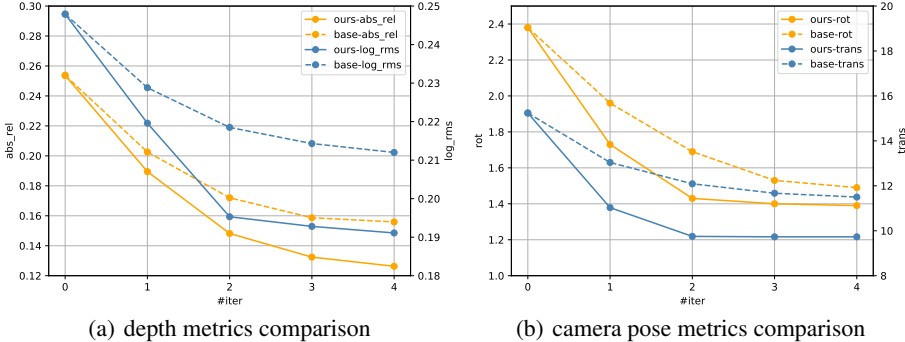

(a) depth metrics comparison  (b) camera pose metrics comparison

Figure 2: Comparison with baseline during iterations. Our work converges at a better position. (a) abs relative error and log RMSE. (b) rotation and translation degree error.

**Iterative Improvement** Our model can run iteratively to reduce the prediction error. Figure 2 (solid lines) shows our performance over iterations when initialized with the prediction from DeMoN. As can be seen, our model effectively reduces both depth and pose errors upon the DeMoN output.

Throughout the iterations, better depth and pose benefit each other by building more accurate cost volume, and both are consistently improved. The whole process is similar to coordinate descent algorithm, and finally converges at iteration 4.

**Effect of P-CV** We compare DeepSFM to a baseline method for our P-CV. In this baseline, the depth prediction is the same as DeepSFM, but the pose prediction network is replaced by a direct visual odometry model Steinbrücker et al. (2011), which updates camera parameters by minimizing pixel-wise photometric error between image features. Both methods are initialized with DeMoN results. As provided in Figure 2, DeepSFM consistently produces lower errors on both depth and pose over all the iterations. This shows that our P-CV predicts more accurate pose and performs more robust against noise depth at early stages.

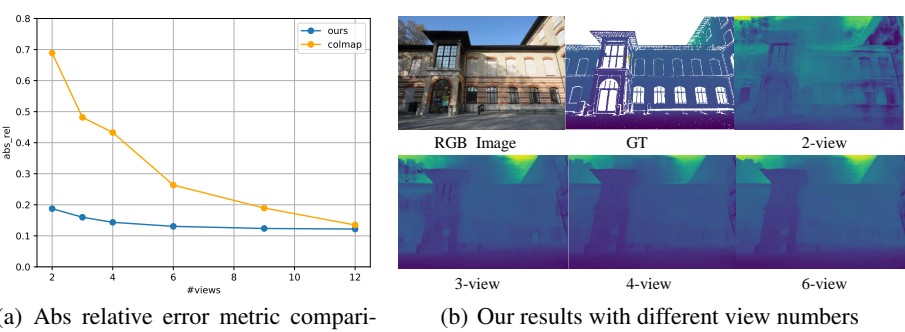

(a) Abs relative error metric comparison

(b) Our results with different view numbers

Figure 3: Depth map results w.r.t. the number of images.

**View Number** DeepSFM works still reasonably well with fewer views due to the free from optimization based components. To show this, we compare to COLMAP with respect to the number of input views on ETH3D. As depicted in Figure 3, more images yield better results for both methods as expected. However, our performance drops significantly slower than COLMAP with fewer number of inputs. Numerically, DeepSFM cuts the depth error by half under the same number of views as COLMAP, or achieves similar error with half number of views required by COLMAP. This clearly demonstrates that DeepSFM is more robust when fewer inputs are available.

## 5    Conclusions

We present a deep learning framework for Structure-from-Motion, which explicitly enforces photometric consistency, geometric consistency and camera motion constraints all in the deep network. This is achieved by two key components - namely D-CV and P-CV. Both cost volumes measure the photo-metric errors and geometric errors but hypothetically move reconstructed scene points (structure) or camera (motion) respectively. Our deep network can be considered as an enhanced learning based BA algorithm, which takes the best benefits from both learnable priors and geometric rules. Consequently, our method outperforms conventional BA and state-of-the-art deep learning based methods for SfM.

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

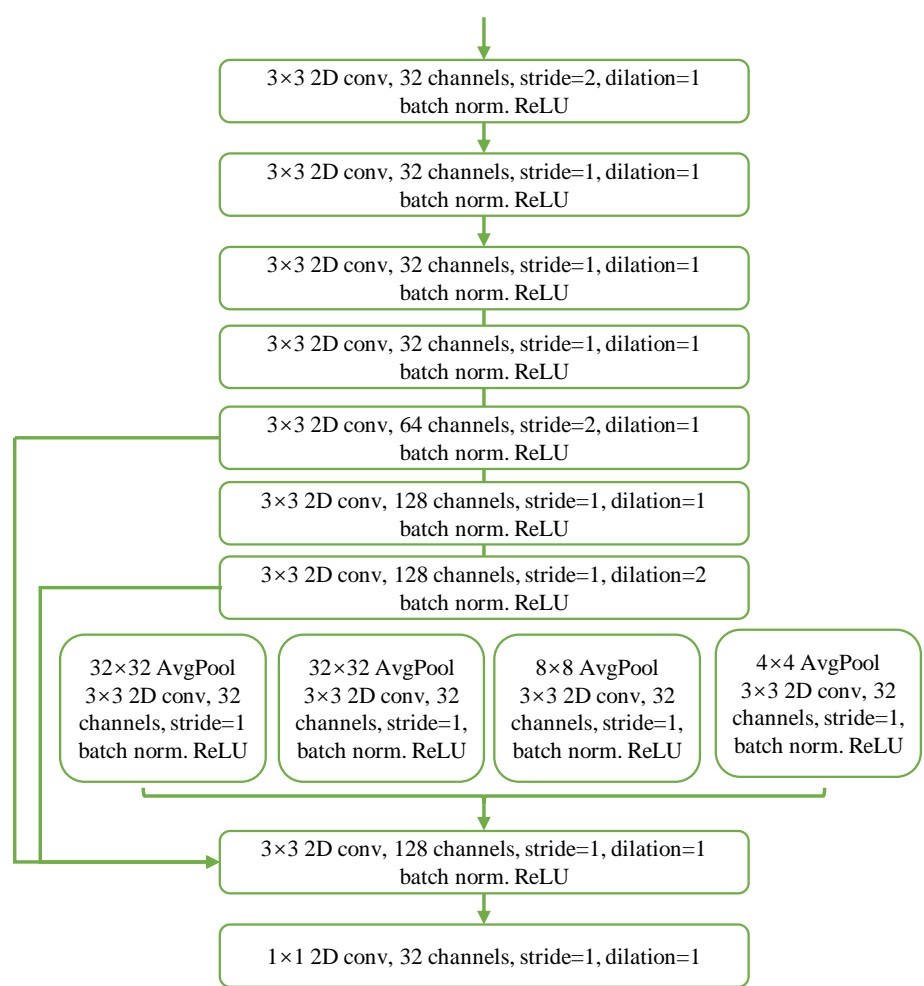

Figure 4: Detail architecture of feature extractor.

## A  IMPLEMENTATION DETAILS

**Feature extraction module**  As shown in Figure 4, we build our feature extraction module refer-ring to DPSNet (Im et al., 2019). The module takes $4W \times 4H \times 3$ images as input and output feature maps of size $W \times H \times 32$, which are used to build D-CV and P-CV.

**Cost volumes**  Figure 5 shows the detailed components for the P-CV and D-CV. Each channel of cost volume is composed of four components: reference view feature maps, warped source view feature maps, the warped source view initial depth map and the projected reference view depth plane or initial depth map. For P-CV construction, we take each sampled hypothetical camera pose, and carry out the warping process on source view depth maps and initial depth map based on the camera pose. And the initial reference view depth map is projected to align numeric values with the warped source view depth map. Finally those four components are concatenated as one channel of 4D P-CV. We do this on all P sampled camera poses, and get the P channel P-CV. The building approach for D-CV is similar, we take each sampled hypothetical depth plane, and carry out warping process on source view feature maps and the initial depth map. And the depth plane is projected to align with the source view depth map. After concatenation, one channel in D-CV is got. Same computation is done based on all L virtual depth planes, and the L channel D-CV is built up.

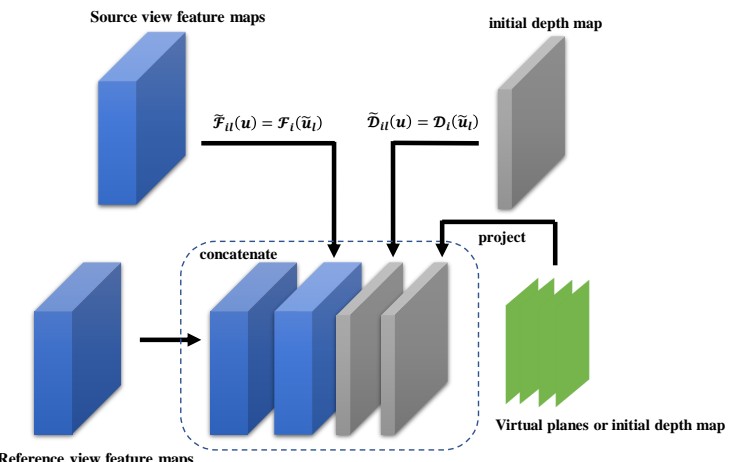

Figure 5: Four components in D-CV or P-CV.

| Method | Depth | | | | | Motion | |
|---|---|---|---|---|---|---|---|
| | abs_rel | sq_rel | rms | log_rms | sc_inv | Rot | Trans |
| Ours | **0.227** | **0.170** | **0.479** | **0.271** | **0.268** | 1.588 | **30.613** |
| BA-Net | 0.238 | 0.176 | 0.488 | 0.279 | 0.276 | **1.587** | 31.005 |
| DeMoN | 0.231 | 0.520 | 0.761 | 0.289 | 0.284 | 3.791 | 31.626 |
| LSD-SLAM | 0.268 | 0.427 | 0.788 | 0.330 | 0.323 | 4.409 | 34.360 |
| Geometric BA | 0.382 | 1.163 | 0.876 | 0.366 | 0.357 | 8.560 | 39.392 |

Table 3: Results on ScanNet. (sc_inv: scale invariant log rms; **Bold**: best.)

**3D convolutional layers**  The detail architecture of 3D convolutional layers after D-CV is almost the same as DPSNet (Im et al., 2019), except for the fist convolution layer. In order to compatible with the newly introduced depth consistent components in D-CV, We adjust the input channel number to 66 instead of 64. As shown in Figure 6, for 3D convolutional layers after P-CV, the architecture is similar to D-CV 3D convolution layers with three extra 3D average pooling layers and finally there is one global average pooling in the dimensions of image width and height, after which we get a $P \times 1 \times 1$ tensor.

## B  EVALUATION ON SCANNET

ScanNet provides a large set of indoor sequences with camera poses and depth maps captured from a commodity RGBD sensor. Following BA-Net, we leverage this dataset to evaluate the generalization capability by training models on DeMoN and testing here. The testing set is the same as BA-Net, which takes 2000 pairs filtered from 100 sequences.

We evaluate the generalization capability of DeepSFM on ScanNet. Table 3 shows the quantitative evaluation results for models trained on DeMoN. The results of BA-Net, DeMoN, LSD-SLAM and Geometric BA are obtained from Tang & Tan (2018). As can be seen, our method significantly outperforms all previous work, which indicates that our model generalizes well to general indoor environments.

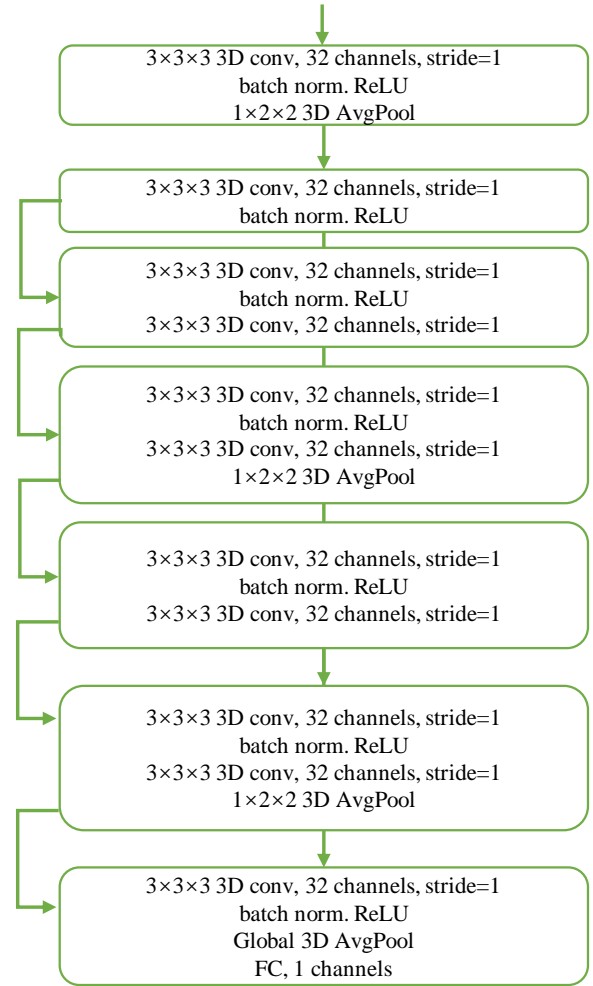

Figure 6: 3D convolutional layers After P-CV.

|             | Initialization | Iteration 2 | Iteration 4 | Iteration 6 | Iteration 10 | Iteration 20 |
|-------------|----------------|-------------|-------------|-------------|--------------|--------------|
| abs relative | 0.254 | 0.153 | 0.126 | 0.121 | 0.120 | 0.120 |
| log rms     | 0.248 | 0.195 | 0.191 | 0.190 | 0.190 | 0.191 |
| translation | 15.20 | 9.75 | 9.73 | 9.73 | 9.73 | 9.73 |
| rotation    | 2.38 | 1.43 | 1.40 | 1.39 | 1.39 | 1.39 |

Table 4: The performance of the optimization iterations for testing.

## C  SUPPLEMENTAL ABLATION STUDY

**More Iterations for Testing.** We take up to four iterations when we train DeepSFM. During inference, the predicted depth maps and camera poses of previous iteration are taken as initialization of next iteration. To show how DeepSFM performs with more iterations than it is trained with, we show results in Table 4. We tested with up to 20 iterations, and it converges at the 6-th iteration.

**Bilinear Interpolation vs Nearest Neighbor Sampling.** For the construction of D-CV and P-CV, depth maps are warped via the nearest neighbor sampling instead of bilinear interpolation. Due to the discontinuity of the depth values in depth maps, the bilinear interpolation may bring some side

| MVS Dataset | L1-inv | sc-inv | L1-rel | Rot | Trans |
|---|---|---|---|---|---|
| Billinear interpolation | 0.023 | 0.134 | 0.079 | 2.867 | 9.910 |
| Nearest neighbor | 0.021 | 0.129 | 0.076 | 2.824 | 9.881 |

Table 5: The performance with different warping methods.

effects. It may do damage to the geometry consistency and smooth the depth boundaries. As a comparison, we replace the nearest neighbor sampling with the bilinear interpolation. As shown in Table 5, the performance of our model gains a slight drop with the bilinear interpolation, which indicates that the nearest neighbor sampling method is indeed more geometrically meaningful for depth. In contrast, the differentiable bilinear interpolation is required for the warping of image features, whose gradients are back propagated to feature extractor layers. Further exploration will be an interesting future work.

## D  VISUALIZATION

We show some qualitative comparison with the previous methods. Since there are no source code available for BA-Net (Tang & Tan, 2018), we compare the visualization results of our method with DeMoN (Ummenhofer et al., 2017) and COLMAP (Schonberger & Frahm, 2016). Figure 7 shows the predicted dense depth map by our method and DeMoN on the DeMoN datasets. As we can see, demon often miss some details in the scene, such as plants, keyboard and table legs. In contrast, our method reconstructs more shape details. Figure 8 shows some estimated results from COLMAP and our method on the ETH3D dataset. As shown in the figure, the outputs from COLMAP are often incomplete, especially in textureless area. On the other hand, our method performs better and always produce an integral depth map. In Figure 9, more qualitative comparisons with COLMAP on challenging materials are provided.

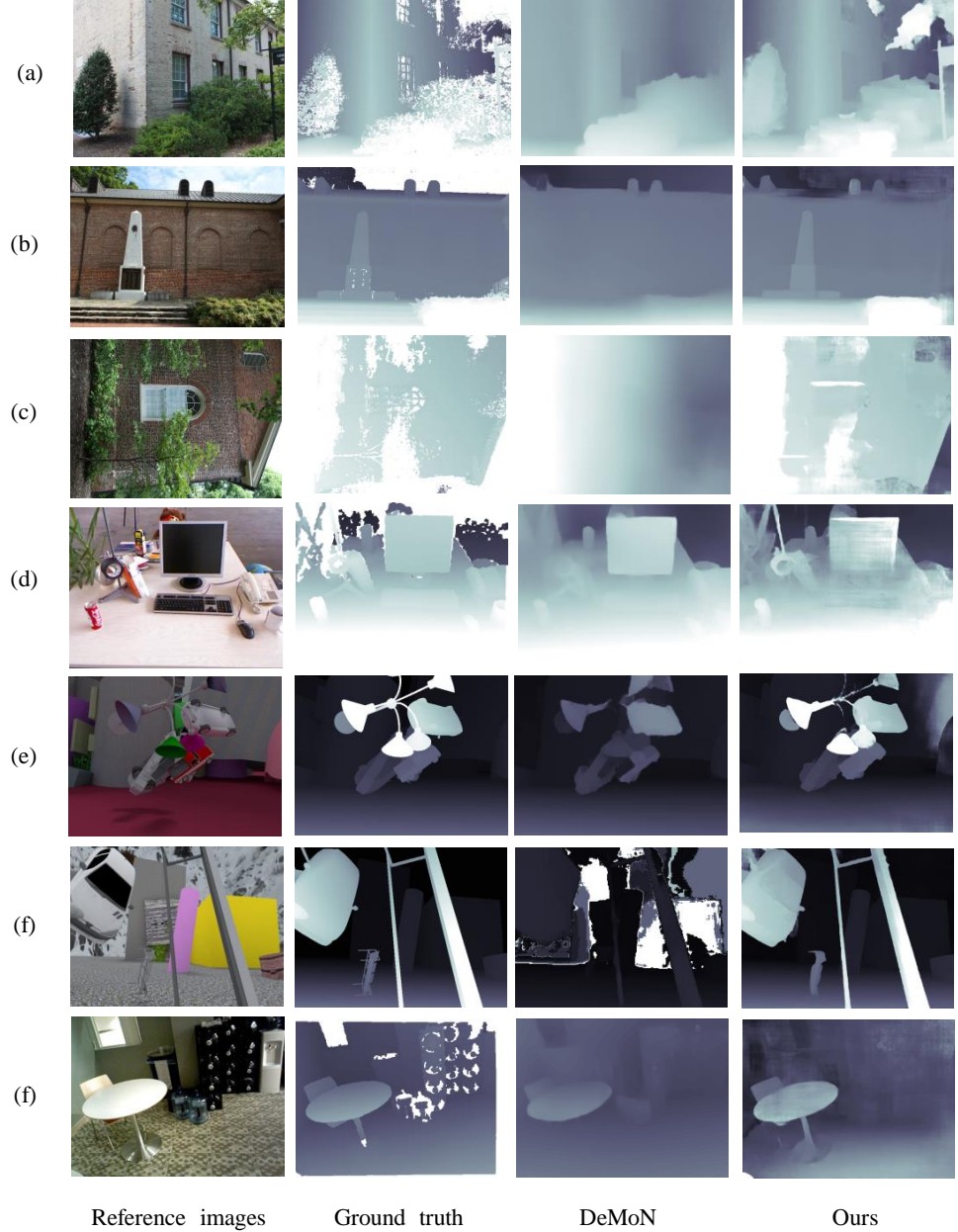

Figure 7: Qualitative Comparisons with DeMoN (Ummenhofer et al., 2017) on DeMoN datasets.

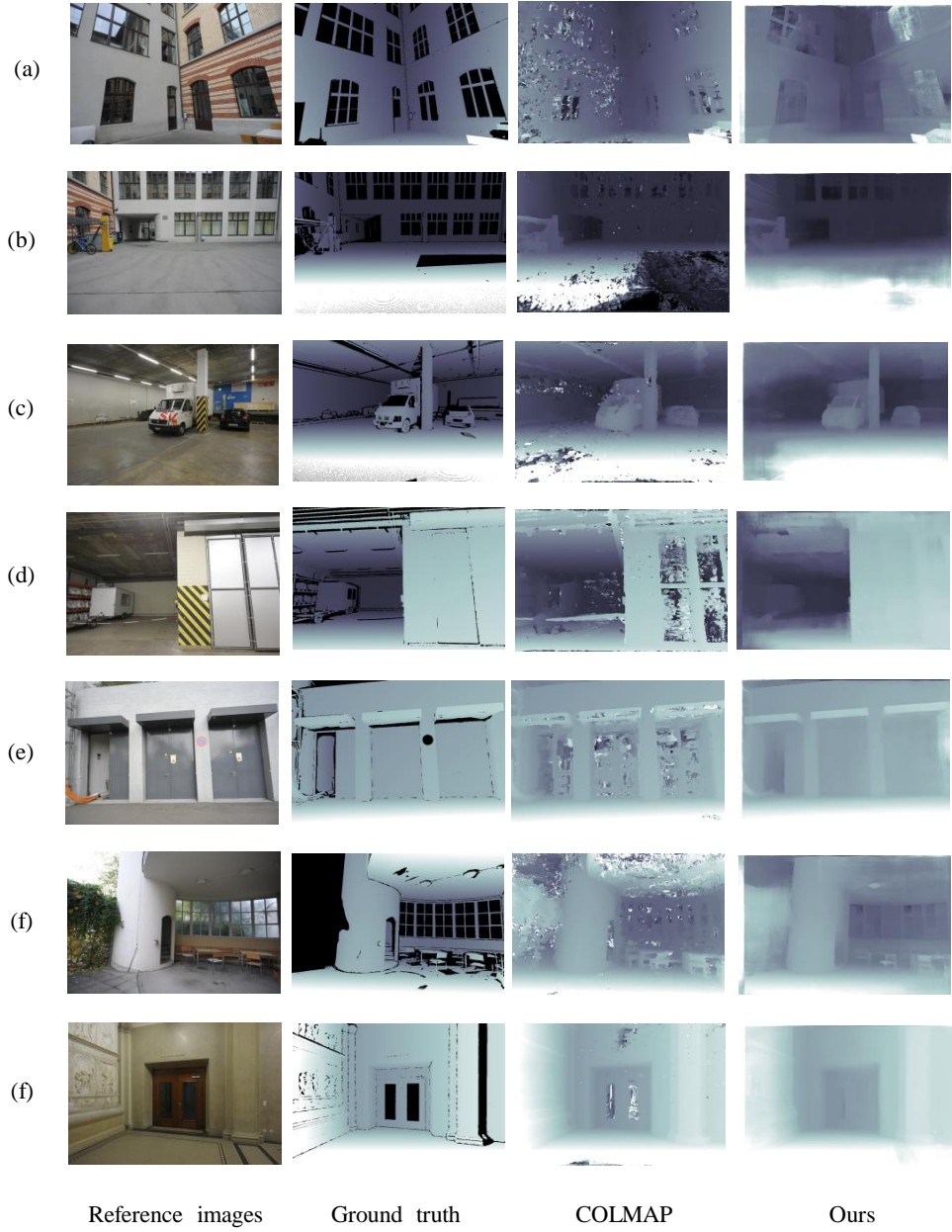

Reference images        Ground truth        COLMAP        Ours

Figure 8: Qualitative Comparisons with COLMAP (Schonberger & Frahm, 2016) on ETH3D datasets.

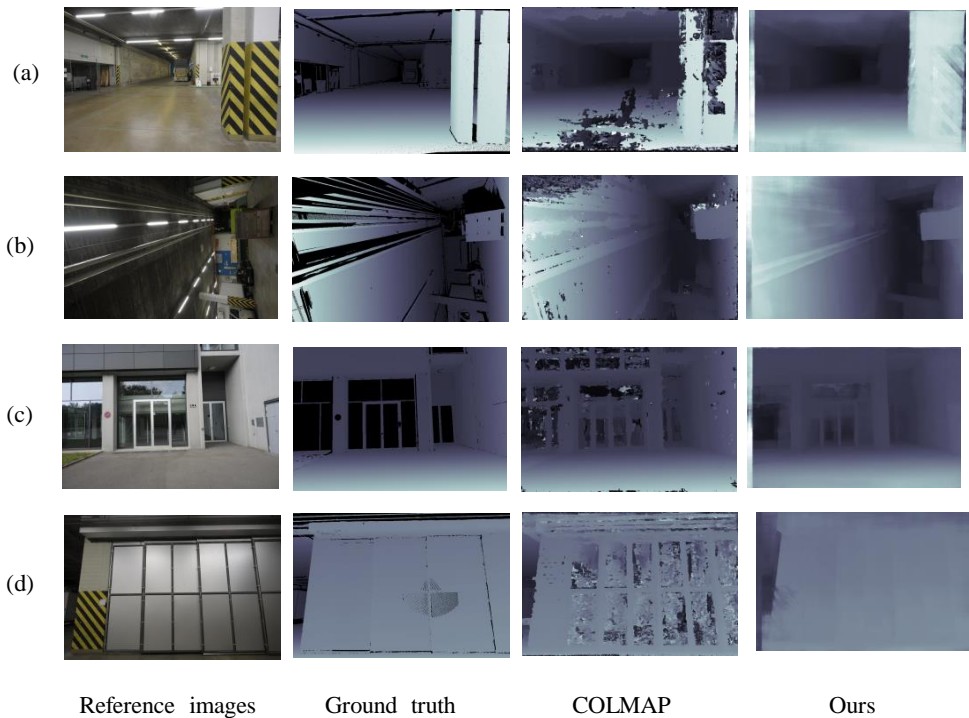

Reference images      Ground truth      COLMAP      Ours

Figure 9: Qualitative Comparisons with COLMAP (Schonberger & Frahm, 2016) on challenging materials. a) Textureless ground and wall. b) Poor illumination scene. c) Reflective and transparent glass wall. d) Reflective and textureless wall.

