# OpenReview forum: "DeepSFM: Structure From Motion Via Deep Bundle Adjustment"
_ICLR.cc/2020/Conference — Reject_

### Official Review · AnonReviewer3 · 2019-10-20
**Official Blind Review #3**

**Rating:** 6

**Review:**

The paper tackles Structure from Motion, one of the canonical problems in computer vision, and proposes an approach that brings together geometry and physics on one hand and deep networks on the other hand. Camera unprojection and warping (of depth maps and features) are used to build a cost volume onto hypothetical planes perpendicular to the camera axis. Similarly, various camera poses are sampled around an initial guess. A deep network regresses form the cost volume to a camera pose and a depth map. The method can be applied iteratively, using the outputs of the current stage as the initial guess of the next one. Training is supervised, and the the results are evaluated on multiple datasets.

I am inclined to recommend accepting the paper for publication, because it addresses a canonical problem, outperforms the state of the art on multiple datasets and brings together geometry / physics and deep learning, which is IMO very a promising and underexplored direction.

I found the method section a bit difficult to read though, and even after several readings I cannot get my head around it. Specifically, here are some issues that I hope the Authors could clarify.

1. In Sec. 3 the Authors write "We then sample the solution space for depth and pose respectively around their initialization". However in Sec 3.2 they write "we uniformly sample a set of L virtual planes {dl} Ll=1 in the inverse-depth space". In what way are the planes "around their initialization"? If the initial depth map spans over multiple orders of magnitude, will the planes be uniformly sampled between the minimum and maximum disparity of the initial map? If yes, it seems that the initial depth map is not really needed, just its minimum and maximum value is needed, but then how come the method can be applied iteratively with respect to depth?

2. The Authors mention that depth maps are warped onto the virtual planes using differentiable bilinear interpolation. Is there a mechanism to protect from interpolating across discontinuities? If no, were bleeding edge artifacts observed?

3. In the introduction, the Authors point that prior methods have trouble dealing with textureless, reflective or transparent approaches, but it's not clear form the paper where it addresses these cases, and if yes, what is the mechanism for that.

Lastly, if the authors are not planning to release the code, the implementation details section is a bit too high-level and does not contain enough details to reimplement the Author's technique. For example, "our network learns a cost volume of size L × W × H using several 3D convolutional layers with kernel size 3 × 3 × 3"  - more details about this network are needed, as well as the others in the paper.



**Experience Assessment:**

I have published one or two papers in this area.

**Review Assessment: Checking Correctness Of Derivations And Theory:**

I assessed the sensibility of the derivations and theory.

**Review Assessment: Checking Correctness Of Experiments:**

I assessed the sensibility of the experiments.

**Review Assessment: Thoroughness In Paper Reading:**

I read the paper at least twice and used my best judgement in assessing the paper.

---

> ### Author Response · Authors · 2019-11-12
> **For Reviewer #3**
>
> We thank the reviewer for the comments and appreciation. We have revised the paper according to the suggestions and would like to clarify as follows:
>
> Q1: In Sec. 3 the Authors write "We then sample the solution space for depth and pose respectively around their initialization". However in Sec 3.2 they write "we uniformly sample a set of L virtual planes {dl} Ll=1 in the inverse-depth space". In what way are the planes "around their initialization"? If the initial depth map spans over multiple orders of magnitude, will the planes be uniformly sampled between the minimum and maximum disparity of the initial map? If yes, it seems that the initial depth map is not really needed, just its minimum and maximum value is needed, but then how come the method can be applied iteratively with respect to depth?
>
> A1. Thank you for pointing this out. "We then sample the solution space for depth and pose respectively around their initialization" is a writing mistake and we have corrected it in our new version. Only the solution space for pose is sampled around initialization. We uniformly sample planes in the inverse-depth(disparity) space between a fixed minimum and maximum range. The initial depth is used for maintaining geometric consistency.
>
> The depth, under such a situation, could still be improved through iterations. Since the pose is improved over the iteration, the depth cost-volume would be updated accordingly, and better depth can be inferred from the more accurate cost-volume.
>
>
> Q2: The Authors mention that depth maps are warped onto the virtual planes using differentiable bilinear interpolation. Is there a mechanism to protect from interpolating across discontinuities? If no, were bleeding edge artifacts observed?
>
> A2. We thank the reviewer for pointing out the potential problem of our warping method on the depth maps. Since depth maps often have discontinuities, we agree with Review #3 that differentiable bilinear interpolation may do damage to the geometry consistency and smooth the edges.  We also updated our experiment results with nearest neighbor instead of bilinear interpolation for depth warping, and revised the corresponding results (Tab. 1-3) and figures in the paper. Notably, our results can get slightly improved by the updated nearest neighbour method inspired by the question asked by Reviewer#3.
>
> To verify this, we added an experiment in Appendix C, which runs nearest neighbor sampling instead of bilinear interpolation. With nearest neighbor warping method, the performance of our model on DeMoN MVS dataset gains a slight boost with retraining. Here are the comparisons:
> MVS dataset			                L1-inv     sc-inv     L1-rel     Rot     Trans
> Ours (bilinear)                               0.023      0.134     0.079     2.867    9.910
> Nearest neighbor(retained)	 0.021	  0.129    0.076     2.824    9.881
>
> This shows that nearest neighbor sampling is indeed more geometrically meaningful for depth. We updated the method to use nearest sampling and update the result accordingly. We also discussed the strengths and weaknesses briefly of each interpolation method in Appendix C.
>
>
> Q3: In the introduction, the Authors point that prior methods have trouble dealing with textureless, reflective or transparent approaches, but it's not clear form the paper where it addresses these cases, and if yes, what is the mechanism for that.
>
> A3. Empirically, learning based method may outperforms traditional feature matching methods on these situations since it relies on image priors. In addition, our method has geometry consistency between multiview depth maps as the input, which encourages local smoothness and consistency to some extent. In some textureless, reflective or transparent cases that feature matching methods does not work, our method gains extra information from the initial depth maps of other views by the depth consistency part of the cost volume. In Appendix D, Figure 8, some qualitative comparisons with COLMAP[1] are provided as an argument. We have updated our paper and show more visual examples in Appendix D, Figure 9.
>
>
> Q4: the implementation details section is a bit too high-level and does not contain enough details to reimplement the Author's technique.
>
> A4. Thanks for your suggestions, we will release code upon the acceptance. Furthermore, we have put more details about model architecture as in Appendix A Figure 4 and Figure 6.
>
> [1] Johannes L Schonberger and Jan-Michael Frahm. Structure-from-motion revisited. In Proceedings of the IEEE Conference on Computer Vision and Pattern Recognition, pp. 4104–4113, 2016.

---

### Official Review · AnonReviewer2 · 2019-10-27
**Official Blind Review #2**

**Rating:** 3

**Review:**

In this paper, the authors propose a physical driven architecture of DeepSFM to infer the structures from motion. Extensive experiments on various datasets show that the model achieves the state-of-the-art performance on both depth and pose estimation. In general, the paper is clearly written but I still have several concerns.
1.	The paper is easy to follow but the authors are expected to clarify the rationality in integration of the loss function. How the parameter of \lambda_r, \lambda_t, and \lambda_r influence the performance. It would be better if the authors could present some analysis.
2.	The experiments are rather insufficient. The authors are expected to make more comprehensive analysis with the state-of-the-art methods, and also analyze why some alternative methods outperforms the proposed methods in table I and table II.
3.	The experiments in section 4.3 are also expected to be improved. It is difficult to draw a conclusion that the method is better than other ones based on such limited experiments.


**Experience Assessment:**

I have published in this field for several years.

**Review Assessment: Checking Correctness Of Derivations And Theory:**

I assessed the sensibility of the derivations and theory.

**Review Assessment: Checking Correctness Of Experiments:**

I assessed the sensibility of the experiments.

**Review Assessment: Thoroughness In Paper Reading:**

I read the paper at least twice and used my best judgement in assessing the paper.

---

> ### Author Response · Authors · 2019-11-12
> **For Reviewer#2**
>
> Thank you very much for your comments, which is very helpful for clarifying our contribution and improving the presentation of the paper. Please see the inline responses.
>
> Q1: The paper is easy to follow but the authors are expected to clarify the rationality in integration of the loss function. How the parameter of \lambda_r, \lambda_t, and \lambda_r influence the performance. It would be better if the authors could present some analysis.
>
> A1: There are in general two rules to follow when choosing the lambda for optimization: 1) the loss term provides gradient in similar numerical range, such that no single loss should dominate the training since accuracies in depth and camera pose are both important to reach a good consensus. 2) we found in practice the camera rotation has higher impact on the accuracy of the depth probably but not the opposite. This is presumably because that pose cost volume accumulate depth differences of all the pixels such that is more tolerant to the depth error. To encourage better performance of pose, we set a relatively large \lambda_r. Note that all the losses are necessary to achieve good performance.  On the validation data, some preliminary experiments by grid search values of each lambda, show that the performance of our model is not very sensitive to various values of lambda. Therefore we provide a combination of lambda that produces the model for our experiment, and presumably there could be other settings that may potentially further improve the performance. We have added some insight to Section 3.5 about the loss function.
>
>
> Q2: The authors are expected to make more comprehensive analysis with the state-of-the-art methods, and also analyze why some alternative methods outperforms the proposed methods in table I and table II.
>
> A2: We thank the reviewer for the suggestion. We add more analysis with the state of the art in Section 4.2, especially about the case that other methods outperforms our method.
>
>
> Q3:The experiments in section 4.3 are also expected to be improved. It is difficult to draw a conclusion that the method is better than other ones based on such limited experiments.
>
> A3: Thanks for this point. However, the main experiments and the conclusions are in Sec. 4.2; and  thus Section 4.2 included  much more insights and discussion of our model Vs. the other baselines in the revised version. In contrast,  Section 4.3 lists the ablation study, where the purpose of experiments is to verify the necessity and sufficiency of some system design options of our model and demonstrate the behavior under controlled experiments, instead of comparing with other methods. Specifically, we show the performance of our method with different number of iterations, with and without pose cost volume, and different numbers of the input view. At the same time, we found that our method also outperforms other methods in some aspects. In Figure 2, the curves are going down, which means that our method can effectively reduce depth and pose error from DeMon. The solid curves are consistently lower than dashed curves, which means our pose cost volume outperforms Steinbrücker et al. (2011) in pose estimation and further benefits depth estimation. In Figure 3, the blue curve is significant lower than orange curve, which means that our method is more robust in the situation with fewer views than COLMAP. Even though, the main purpose is not to compare to others but provide some analysis on important model components.

---

### Official Review · AnonReviewer1 · 2019-11-08
**Official Blind Review #1**

**Rating:** 6

**Review:**

Summary:

The authors propose a SfM model which integrates geometric consistency with a learned pose and depth network. An initial estimate of depth and pose are used to construct pose and depth cost volumes, which are then fed into a pose regression and depth refinement network, to produce a new set of cost volumes, and so on. In this manner, the pose and depth estimation are improved iteratively.

Strengths:

The proposed model is well motivated and shows strong performance and generalization ability on several datasets. There are convincing experiments to show the importance of the P-CV network.

Weaknesses:

The authors claim that the LM optimization in BA-Net is memory inefficient and may lead to non-optimal solutions. It’s not clear to me that the proposed method can guarantee optimality any better. It’s also unclear if the proposed method is more memory efficient, since the authors only unroll 4 iterations of it.

Other comments:

It would be very interesting to see the test time behavior of the network when it is run with more iterations than it is trained with (say 10 or 20), especially since the depth error does not seem to have stopped decreasing at only 4 iterations.

It’s not made entirely clear whether the training backpropagates through the update/construction of the pose and depth cost volumes.

In equation 5, “x” should be “i”.


**Experience Assessment:**

I have published in this field for several years.

**Review Assessment: Checking Correctness Of Derivations And Theory:**

I assessed the sensibility of the derivations and theory.

**Review Assessment: Checking Correctness Of Experiments:**

I carefully checked the experiments.

**Review Assessment: Thoroughness In Paper Reading:**

I read the paper thoroughly.

---

> ### Author Response · Authors · 2019-11-12
> **For Reviewer #1**
>
> We thank the reviewer for the comments and appreciation, and would like to answer the reviewer’s questions as follows:
>
> Q1:The authors claim that the LM optimization in BA-Net is memory inefficient and may lead to non-optimal solutions. It’s not clear to me that the proposed method can guarantee optimality any better. It’s also unclear if the proposed method is more memory efficient, since the authors only unroll 4 iterations of it.
>
> A1: Thanks for pointing this out and sorry for the confusion! Here we don’t mean that our method can fix the optimality problem in any way. We wish to provide some of our analysis of the limitation of BA-Net, and hope our method could provide complementary perspectives to rethink the problem and mitigate the non-optimal issue in terms of performance with more ML component. In terms of number of iterations, our method does not have a restriction, since our iteration happens outside the neural network and acts as an incremental improvement. In contrast, BA_Net’s iteration is part of the LM optimization and it is inside the network. Thus if it unrolls more iteration steps, the memory cost will increase linearly. We have updated the paper for this.
>
>
> Q2: Show the test time behavior of the network when it is run with more iterations than it is trained with (say 10 or 20)
>
> A2: Thanks for the suggestion! We added Table 4 in Appendix C that shows performance of the network with more iterations(from 2 to 20).
>
>
> Q3:It’s not made entirely clear whether the training back propagates through the update/construction of the pose and depth cost volumes.
>
> A3: Gradients can back-propagate through cost volumes, and cost-volume construction does not affect any trainable parameters. We updated this point in the revised version.
>
>
> Q4: In equation 5, “x” should be “i”.
>
> A4: Thanks for pointing out that! We have fixed the typo.

---

### Author Response · Authors · 2019-11-12
**Summarization of changes in our new version**

We thank all the reviewers for their insightful and constructive comments. We have revised the paper as suggested by the reviewers, and summarize the major changes as follows:
1， In Introduction, we rewrote the sentences that discuss the LM optimization in BA-Net.
2， In Page 3 Section 3 paragraph 2, We fixed a writing mistake. “We then sample the solution space for depth and pose respectively around their initialization” -> “We then sample the solution space for depth uniformly in the inverse-depth space between a predefined minimum and maximum range and camera pose around the initialization respectively.“ We thank review #3 for pointing it out.
3， In section 3.5, We added some insight to explain the rationality in integration of the loss function as required by review #2.
4，We fixed the error in equation 5: x -> i. Thanks to review #1 for pointing it out.
5， In section 4, We updated our experiment results with nearest neighbor instead of bilinear interpolation for depth warping. We also added a sentence in section 3.2 which clarifies that we adopt nearest neighbor sampling for depth warping, instead of bilinear interpolation. Thanks to review # 3 for pointing the weakness of bilinear interpolation out.
6， Section 4.2 was extended to include brief introduction of the state-of-art methods we compared with. In addition, more analysis required by review #2 about the results in table I and table II were added.
7， In Appendix A, We added Figure 4 and Figure 6 which shows more details  required by review #3 about the implementation.
8， In Appendix C, We added Table 4 required by review #1 which shows the performance with more number (up to 20) of iterations.
9， In Appendix C, we added an experiment that compares Bilinear Interpolation with Nearest Neighbor as the answer to review #3.
10，In Appendix D. We added figure 9 which shows qualitative comparisons with COLMAP (Schonberger & Frahm, 2016) on challenging materials, as a supplemental answer to review #3.

---

### Decision · Program_Chairs · 2019-12-19

**Decision:**

Reject

**Comment:**

Main content:  Physical driven architecture of DeepSFM to infer the structures from motion
Discussion:
reviewer 1: well-motivated model with good solid experimental results. not clear about the LM optimization in BA-Net is memory inefficient
reviewer 2: main issue is the experiments could be improved.
reviewer 3: well written but again experimental section is lacking
Recommendation: Good paper and results, but all 3 reviewers agree experiments could be improved. Rejection is recommended.